# Anlotinib Exerts Inhibitory Effects against Cisplatin-Resistant Ovarian Cancer In Vitro and In Vivo

**DOI:** 10.3390/molecules27248873

**Published:** 2022-12-14

**Authors:** Yurou Ji, Xinyu Li, Yue Qi, Jianguo Zhao, Wenwen Zhang, Pengpeng Qu

**Affiliations:** 1Clinical School of Obstetrics and Gynecology Center, Tianjin Medical University, Tianjin 300100, China; 2Tianjin Central Hospital of Obstetrics and Gynecology, Tianjin 300100, China

**Keywords:** ovarian cancer, cisplatin-resistant, anlotinib

## Abstract

Background: Anlotinib is a highly potent multi-target tyrosine kinase inhibitor. Accumulating evidence suggests that anlotinib exhibits effective anti-tumor activity against various cancer subtypes. However, the effects of anlotinib against cisplatin-resistant (CIS) ovarian cancer (OC) are yet to be elucidated. The objective of this study was to investigate the inhibitory effect of anlotinib on the pathogenesis of cisplatin-resistant OC. Materials and Methods: Human OC cell lines (A2780 and A2780 CIS) were cultured and treated with or without anlotinib. The effects of anlotinib on cell proliferation were determined using cell-counting kit-8 and colony-formation assays. To evaluate the invasion and metastasis of OC cells, we performed wound-healing and transwell assays. The cell cycle was analyzed via flow cytometry. A xenograft mouse model was used to conduct in vivo studies to verify the effects of anlotinib. The expression of Ki-67 in the tumor tissue was detected via immunohistochemistry. Quantitative real-time polymerase chain reaction and Western blotting were used to measure the mRNA and protein levels. Results: Our study revealed that anlotinib significantly inhibited the proliferation, migration, and invasion of A2780 and A2780 CIS in a dose-dependent way in vitro (*p* < 0.05). Through R software ‘limma’ package analysis of GSE15372, it was found that, in comparison with A2780, PLK2 was expressed in significantly low levels in the corresponding cisplatin-resistant strains. The ERK1/2/Plk2 signaling axis mediates the inhibitory effect of anlotinib on the proliferation and migration of ovarian cancer cell lines. Moreover, our research found that anlotinib effectively inhibited the growth of tumor cells in an OC xenograft mouse model. Conclusions: In this study, anlotinib showed excellent inhibitory effects against cisplatin-resistant OC both in vitro and in vivo. These results add to the growing body of evidence supporting anlotinib as a potential anticancer agent against OC.

## 1. Introduction

Ovarian cancer (OC) is one of the most common cancers diagnosed in females [1,2]. According to global cancer statistics published in 2020, more than 313,000 females are diagnosed with OC and 207,000 die every year globally [2,3]. Owing to the absence of early symptoms and effective early-stage detection strategies, more than 70% of patients with OC are diagnosed at an advanced stage. Surgery and platinum-based cytotoxic chemotherapy are still considered front-line standard therapy [4]. However, due to the development of drug resistance, particularly resistance to platinum, the majority of patients—70 to 80% of them—experience a relapse within two years of completion of first-line therapy [5,6]. The development of platinum resistance continues to be a significant clinical challenge [7]. Hence, there is an urgent need to explore novel treatments and improve the prognosis of platinum-resistant OC. 

Targeted therapy has recently shown potential for the treatment of malignant tumors. Accumulating evidence suggests that multi-target tyrosine kinase inhibitors (TKIs), including vascular endothelial growth factor (VEGF) receptor inhibitors, such as nintedanib and apatinib, improve outcomes among patients with platinum-resistant OC [8,9]. Anlotinib is a new oral TKI that targets VEGFR2/3, platelet-derived growth factor receptor α/β, fibroblast growth factor receptor 1–4, MET, stem cell factor receptor, and RET [10,11,12,13,14]. Numerous domestic and international studies have found that this drug has four major functions: anti-tumor angiogenesis [12,14]; anti-tumor lymphangiogenesis [15]; inhibition of tumor growth [16,17]; and remodeling the tumor microenvironment [18,19,20,21]. It can also inhibit the development of solid tumors through various mechanisms. To date, anlotinib has shown promising results in the management of various carcinomas. Anlotinib plays a critical anti-tumor role of the epithelial–stroma interaction, mediated by lactate/BDNF/TRKB signaling in gastric cancer resistance [22] and the anti-tumor activity of anlotinib in oral squamous cell carcinoma involved in the suppression of mitochondrial respiration via NOX5-mediated redox imbalance and the AKT/eIF2α pathway [23]. Recently, case reports and clinical trials have demonstrated that patients with platinum-resistant OC respond well to anlotinib [24,25], thereby indicating its potential utility for patients with platinum-resistant OC. However, no preclinical studies have investigated the efficacy of anlotinib against cisplatin-resistant (CIS) OC or explored its possible molecular mechanisms. Here, we aimed to evaluate whether anlotinib exerts any effect against resistant OC under in vitro and in vivo conditions and attempted to explore the underlying mechanisms.

## 2. Results

### 2.1. Anlotinib Inhibits Proliferation, Migration, and Invasion of OC Cells In Vitro

To determine the inhibitory effect of anlotinib on OC cells, we first used the cell-counting kit 8(CCK-8) assay to assess the antiproliferation effects of the drug. The OC cell line A2780 was incubated with anlotinib for 72 h. The viability of A2780 cells was significantly inhibited in a dose-dependent manner (Figure 1A,B). Colony-formation assays were performed to confirm the effect of anlotinib on A2780 cell proliferation (Figure 1C,D). Given the antiproliferative effect of anlotinib on OC cells, we explored whether anlotinib inhibits cell proliferation by regulating cell-cycle progression. The A2780 cells were treated with or without anlotinib. The cells were then stained with PI and analyzed via flow cytometry (FCM). The fraction of G2/M cells increased markedly in the anlotinib-treated group (Appendix A). These results indicate that anlotinib partly inhibited proliferation by arresting the cell cycle in the G2/M phase. In addition, we performed wound-healing and transwell assays to evaluate whether anlotinib inhibited OC cell migration and invasion. The transwell assay revealed that anlotinib inhibited the invasive ability of the A2780 cells (Figure 1E,F). The wound-healing assay results indicated that the migration of A2780 cells progressively decreased with increasing concentrations of anlotinib (Figure 1G,H). These data suggest that anlotinib significantly inhibited the proliferation, migration, and invasion of ovarian cells. 

### 2.2. Anlotinib Inhibits Proliferation, Migration, and Invasion of Cisplatin-Resistant OC Cells In Vitro

Given the anti-tumor effect of anlotinib on OC cells, we hypothesized that anlotinib would also inhibit the proliferation of drug-resistant OC cells. Without drug interference, the growth rate of cisplatin-resistant A2780 (A2780 CIS) was significantly higher than that of sensitive cell lines (Figure 2A). However, the growth of A2780 CIS cells was significantly inhibited by anlotinib (Figure 2B). Colony-formation assays also indicated that anlotinib remarkably slowed the growth of the A2780 CIS cells (Figure 2C,D). Moreover, anlotinib induced G2/M arrest in cisplatin-resistant OC cells (Appendix A). As anlotinib had significant effects on the migration and invasion of A2780 cells, we speculated that anlotinib might repress cisplatin-resistant OC cell migration and invasion. To corroborate this hypothesis, wound-healing assays were performed. A2780 CIS cells treated with anlotinib showed markedly slower migration than the control group (Figure 2E,F). Additionally, A2780 CIS cells treated with anlotinib showed a substantially decreased invasion ability (Figure 2G,H). These results imply that anlotinib inhibited the proliferation, migration, and invasion of cisplatin-resistant OC cells.

### 2.3. Anlotinib Inhibits the Proliferation of Cisplatin-Resistant OC Cells by Inducing the Expression of PLK2

The downregulation of PLK2 expression has been linked to ovarian tumorigenesis and drug resistance [26]. The Gene Expression Omnibus (GEO) dataset GSE15372 was used to identify PLK2 as a differentially expressed gene (DEG) for drug resistance in ovarian cancer (Figure 3A), and Western blotting and qRT-PCR showed that PLK2 expression was significantly lower in drug-resistant ovarian cells (Figure 3B–D). In the present study, we explored the expression of PLK2 in cisplatin-resistant ovarian cancer cells treated with or without anlotinib. The results showed that the expression of PLK2 was upregulated at the protein and mRNA levels (Figure 3E–G), indicating that anlotinib may reduce some of the resistance by upregulating the expression of PLK2. To investigate whether anlotinib inhibited the proliferation of cisplatin-resistant ovarian cancer cells by inducing the expression of PLK2, we constructed si-PLK2 cells. The knockdown efficiency was verified by Western blotting and qRT-PCR (Figure 3H–J). PLK2 knockdown accelerated cell growth. In contrast, anlotinib inhibited cell growth. However, this effect was attenuated by PLK2 knockdown (Figure 3K), indicating that PLK2 was involved in the anti-tumor effect of anlotinib.

### 2.4. Potential Molecular Mechanisms Underlying the Inhibition of the Growth of Drug-Resistant OC Mediated by Anlotinib

GSE15372 differential gene enrichment analysis revealed the enrichment of the MAPK pathway (Figure 4A). To explore the molecular mechanisms underlying anlotinib-induced tumor inhibition, we investigated the activity of ERK/MAPK via Western blotting. ERK1/2 phosphorylation was suppressed by anlotinib in a dose-dependent manner, and a marked suppression was observed at a dose of 8 µM in both cell lines (Figure 4B,C). These results indicate the possible anti-tumor molecular mechanisms of anlotinib.

### 2.5. Anlotinib Suppresses Tumour Growth in Mice Harbouring Xenografts of Cisplatin-Resistant OC Cells

To examine the therapeutic significance of anlotinib in vivo, a tumor xenograft mouse model was established by injecting A2780 CIS cells into the flanks of nude mice. Anlotinib significantly suppressed the growth of cisplatin-resistant ovarian tumors compared to that observed in the control group (*p* < 0.05) (Figure 5A–D), without affecting the bodyweight of the mice. Consistent with this finding, the post-mortem tumor weight in the anlotinib treatment group was significantly lower than that in the control group (*p* < 0.05). Subsequently, the proliferation of cancer cells was observed using Ki-67 immunohistochemical staining to determine the curative efficacy of anlotinib on tumor growth. The results indicate that Ki67 expression was clearly downregulated in the groups treated with anlotinib, indicating fewer proliferating cells (Figure 5E). To further explore the underlying mechanism by which anlotinib inhibited cisplatin-resistant ovarian tumor growth in vivo, the expression of PLK2 was quantified via qRT-PCR and Western blotting (Figure 5F,G). PLK2 levels were markedly elevated in the anlotinib-treated group. These results are consistent with the data obtained from the in vitro experiments. These data demonstrate that anlotinib inhibited cisplatin-resistant ovarian tumor growth partly by increasing the expression level of PLK2 in vivo.

## 3. Discussion

OC is the most lethal gynecological cancer with a five-year survival rate < 45%. Chemotherapy plays a major role in both adjuvant treatment and in the care of patients with advanced-stage OC [27]. Cisplatin is the first-line chemotherapeutic agent for the treatment of OC, and its adverse effects and resistance against the drug observed in tumors are major barriers to successful OC treatment [28]. Therefore, studies aimed at developing novel therapies that increase survival rates associated with cisplatin-resistant OC are urgently needed. Drugs targeting molecules involved in the pathogenesis and progression of cancer have emerged as potentially suitable therapeutic agents. 

Anlotinib is a novel oral small molecule, multi-target TKI that has been reported to exert anti-tumor effects against many types of cancers in recent preclinical studies and ongoing clinical trials. Anlotinib induces hepatocellular carcinoma apoptosis and inhibits proliferation via the ERK and AKT pathway [29], and it was reported that anlotinib could reverse chemotherapy resistance in osteosarcoma [30]. Moreover, according to a retrospective observational study, anlotinib showed moderate improvements in progression-free survival and overall survival in patients with platinum-resistant or platinum-refractory OC [31]. However, the effect of anlotinib on cisplatin-resistant cells in OC has rarely been reported. In our study, we demonstrated that anlotinib exerted potent anti-tumor effects in different preclinical cisplatin-resistant models. Using in vivo models where a tumor xenograft mouse model was established, we found that anlotinib significantly suppressed the growth of cisplatin-resistant ovarian tumors compared to that observed in the control group by increasing the expression level of PLK2. In vitro, anlotinib inhibits the proliferation, migration, and invasion of cisplatin-resistant OC cells. The results of this study provide compelling data regarding anlotinib function and how it impacts the proliferation, migration, and invasion of cisplatin-resistant OC cells.

Here, we found that anlotinib increased the expression level of PLK2. The downregulation of PLK2 expression is associated with ovarian tumorigenesis and drug resistance [26]. For instance, in a PLK2 knock-in and knockdown model, constructed using a primary cell culture, it was confirmed that therapeutic drug resistance was related to the transcriptional silencing of the kinase. This finding is in accordance with the significant downregulation of PLK2 identified in chemo-resistant ovarian cancer cells via oligonucleotide microarrays [26,32,33,34]. In our study, higher PLK2 expression indicated a stronger anti-tumor effect. Hence, PLK2 might be a target of anlotinib for the inhibition of cisplatin-resistant OC. However, the specific mechanism requires further investigation. Several studies have shown that abnormal activation of the ERK/MAPK signaling pathway can promote drug resistance in tumors [35]. ERK is associated with cisplatin resistance in OC [36,37,38,39,40]. The findings of the present study suggest that anlotinib partially suppresses cell proliferation via the inhibition of the MAPK/ERK signaling pathway in cisplatin-resistant OC and could be a novel therapeutic strategy for the treatment of OC.

Based on the findings of the current study, we revealed the anti-tumorigenic effect of anlotinib in cisplatin-resistant OC cells. These results will be helpful in clarifying the potential inhibitory effects of anlotinib on cisplatin-resistant ovarian cancer progression. Further studies examining the effects of anlotinib in patients with OC are warranted. The long-term toxicity of anlotinib in OC requires further analysis. Although our study has limitations, we are now planning studies to validate our findings on the role of anlotinib in cisplatin-resistant OC cells using PLK2 knockout mice. Such studies will allow us to better describe the underlying molecular mechanisms by which anlotinib inhibits cisplatin-resistant OC.

## 4. Materials and Methods

### 4.1. Drugs and Antibodies

Anlotinib was obtained from Chia Tai Tianqing Pharmaceutical Group Co., Ltd. and dissolved in saline. A 10 mM stock solution was prepared and stored. Diluted solutions of all reagents were freshly prepared before each experiment. Antibodies against PLK2 were purchased from Cell Signaling Technology, Inc. (Danvers, MA, USA). Antibodies against piERK1/2 (p44/42) and Erk1/2 were purchased from Cell Signaling Technology, Inc. (Danvers, MA, USA). Primary antibodies against GAPDH, β-actin, and secondary antibodies were purchased from ABclone (Wuhan, China).

### 4.2. Cell Culture

The human OC cell line A2780 was purchased from the American Type Culture Collection (ATCC, Rockville, MD, USA). The A2780 CIS cell line was a kind gift from Professor Bin Li of Cancer Hospital Chinese Academy of Medical Sciences. All cells were cultured in the RPMI-1640 medium. The cells were maintained in a medium containing 10% (*v*/*v*) fetal bovine serum (HyClone, Logan, UT, USA) and 1% penicillin/streptomycin (Gibico, Shanghai, China). All cells were incubated at 37 °C in an incubator with 5% CO_2_. Cisplatin resistance among A2780 CIS cells was detected via the CCK-8 assay (Figure 2A).

### 4.3. Cell Viability Assay

CCK-8 (Beyotime, Nantong, China) was used to analyze cell viability following treatment with anlotinib, according to the manufacturer’s instructions. Briefly, 100 μL of cell suspension from each subgroup obtained at a density of 4 × 103 cells/well was seeded in a 96-well plate for 24 h, and then the cells were treated with anlotinib (0, 1, 2, 4, 8, and 10 µM) for 72 h at 37 °C. Subsequently, a 10% CCK-8 solution (the ratio of the volume of the medium to CCK-8 was 9:1) was added to each well, followed by incubation for 1 h. Following further incubation for 1 h at 37 °C in the dark, the absorbance was measured at 450 nm using a microplate reader (Molecular Devices, LLC, San Jose, CA, USA). The cell viability rate was calculated using the following formula: (experimental absorbance − background absorbance)/(control absorbance − background absorbance) × 100%.

### 4.4. Colony-Formation Assay

Five hundred cells were seeded into six-well plates and treated with or without the indicated concentrations of anlotinib. The medium was changed every three days. After incubation for two weeks, the colonies were fixed with 4% paraformaldehyde for 20 min and stained with 0.2% crystal violet for 10 min. The formative colonies were photographed and counted using ImageJ.

### 4.5. Wound-Healing Assay

A wound-healing assay was performed to assess cell migration. Briefly, 2 × 105 cells were individually seeded in six-well plates and placed in an incubator with 5% CO_2_ at 37 °C. After the cells reached 80% confluence, they were treated with (4 or 8 µM) or without anlotinib at 37 °C and cultured in the medium for 24 h. A linear wound was created by scraping the confluent cell monolayer with a 10 μL pipette tip. The cells were washed thrice with phosphate-buffered saline (PBS) and cultured in a fresh, serum-free medium for an additional 24 h. The images of cell migration were captured at 0, 12, and 24 h at the same location.

### 4.6. Transwell Assay

For the transwell invasion assay, 5 × 10^4^ cells were trypsinized and seeded onto the upper chamber coated with Matrigel containing 100 μL of a serum-free medium. The lower chambers were supplemented with 600 μL of a medium containing 10% fetal bovine serum. The non-invasive cells in the membrane of the upper surface were removed with a cotton tip after 24 h of incubation at 37 °C, while the cells on the lower surface that invaded through the pores were fixed with 4% paraformaldehyde for 15 min, stained with 0.1% crystal violet for 30 min, and then washed with PBS. The number of invasive cells was observed from the digital images captured using a microscope and were calculated using the ImageJ software.

### 4.7. Cell-Cycle Assay

Briefly, the cells were digested with trypsin and centrifuged at 300× *g* for 5 min. After washing twice with PBS, the cells were fixed in 70% ethanol at −20 °C overnight. The cells were incubated with propidium iodide (PI; 500 μL) in the dark for 10 min at 37 °C. Finally, the cell cycle was evaluated using FCM (BD Biosciences, San Jose, CA, USA) and analyzed using FlowJo software.

### 4.8. Bioinformatic Analysis

The OC data analyzed in this study were obtained from GSE15372. The DEGs were identified in the two groups of patients using the limma R package. Gene enrichment analysis was performed using the clusterProfiler R package.

### 4.9. Western Blotting

First, the cells were lysed with a cold RIPA buffer (Solarbio, Beijing, China) and concentrations of proteins were quantified using a BCA protein quantification kit (Invitrogen, Thermo Scientific, Waltham, MA, USA). A total of 40 µg of the protein sample was subjected to sodium dodecyl sulphate–polyacrylamide gel electrophoresis. Proteins were transferred onto poly (vinylidene fluoride) membranes (Millipore, Merck, Shanghai, China), which were then blocked with 5% non-fat milk for 1 h. The membranes were incubated with primary antibodies overnight at 4 °C. After washing thrice with TBST, the membrane was incubated with a secondary antibody at room temperature for 1 h. The target protein bands were detected using a chemiluminescent analyzer (GE, USA) with an enhanced chemiluminescent horseradish peroxidase substrate (Millipore, MA, USA), and GAPDH or β-actin was used as a loading control. The films were then quantified using Image J software.

### 4.10. Quantitative Real-Time Polymerase Chain Reaction (qRT-PCR)

Total RNA was isolated from cultured cell lines using a TRIzol reagent, and reverse transcription was performed to obtain cDNA using a reverse transcription kit (Mei5bio, Beijing, China) according to the manufacturer’s instructions. qRT-PCR analyses were performed with a 20 μL mixture comprising 2 × SYBR Green qRT-PCR Master Mix (Bimake, Shanghai, China) and gene-specific oligonucleotide primers (PLK2, 5′-forward primer: CTACGCCGCAAAAATTATTCCTC, reverse primer: 5′-TCTTTGTCCTCGAAGTAGTGGT; GAPDH, forward primer: 5′-AGAAGGCTGGGGCTCATTTG; reverse primer: 5′-AGGGGCCATCCACAGTCTTC). In the analysis of the data, the relative expression of genes was expressed as 2−ΔΔCt (ΔCt = Ct targeted gene—Ct internal gene; −ΔΔCt = ΔCt control group—ΔCt experimental group).

### 4.11. Xenograft Model

Four-week-old female BALB/c nude mice were purchased from Tianjin Animal Center (Tianjin, China) and housed under the following conditions: at a specific-pathogen-free (SPF) animal facility with ad libitum access to food and water. The xenograft mouse model was constructed as follows: Briefly, A2780 CIS cells (100 μL, 1 × 106) were subcutaneously inoculated in the flank of BALB/c nude mice. When the tumor volume reached approximately 100 mm^3^, the mice (*n* = 10) were randomly divided into control and anlotinib groups. Mice in the anlotinib group were treated with 3 mg/kg/day of anlotinib, and mice in the control group were treated with an equivalent volume of saline for 14 days. The drugs in the control and anlotinib groups were administered via intraperitoneal (i.p.) injection. The tumor volume was calculated daily using the following formula: volume = length × width2/2. Finally, the mice were euthanized on day 14 after injection, and the size and weight of the tumors were evaluated. The animal experiments in this study were approved by the Animal Ethics Committee of Tianjin Medical University.

### 4.12. Haematoxylin and Eosin (HE) Staining Immunohistochemistry

Tumor tissues were embedded in paraffin and cut into 3 mm sections for staining. For histopathological assessment, the tissue sections were stained with an HE staining kit after they were deparaffinized with dimethylbenzene. For evaluation via immunohistochemistry, the samples were incubated with a primary antibody against Ki-67 (1:250; Cell Signaling Technology Inc.; Danvers, MA, USA) overnight at 4 °C and then incubated with a horseradish-peroxidase-conjugated secondary antibody (1:500; ZSGB-BIO, Beijing, China) at room temperature for 2 h. Positive cells in the tumor tissue sections were observed and photographed using a light microscope.

### 4.13. Statistical Analysis

SPSS version 26.0 (IBM Corporation, Armonk, NY, USA) was used for data analyses. All experiments were performed in at least three biological replicates, and each biological replicate contained three technical replicates. Differences between groups were analyzed using two-tailed Student’s *t*-tests and analysis of variance. Values of * *p* < 0.05, and ** *p* < 0.01 were considered statistically significant. Results are presented as the mean ± standard error of the mean.

## 5. Conclusions

In summary, this study provides robust evidence supporting the anti-tumor effects of anlotinib in cisplatin-resistant OC. In vitro and in vivo experiments showed that anlotinib inhibited cell proliferation, migration, and invasion, and arrested the cells in the G2/M phase. The results indicate that anlotinib is a potential anticancer agent for cisplatin-resistant OC therapy.

## Figures and Tables

**Figure 1 molecules-27-08873-f001:**
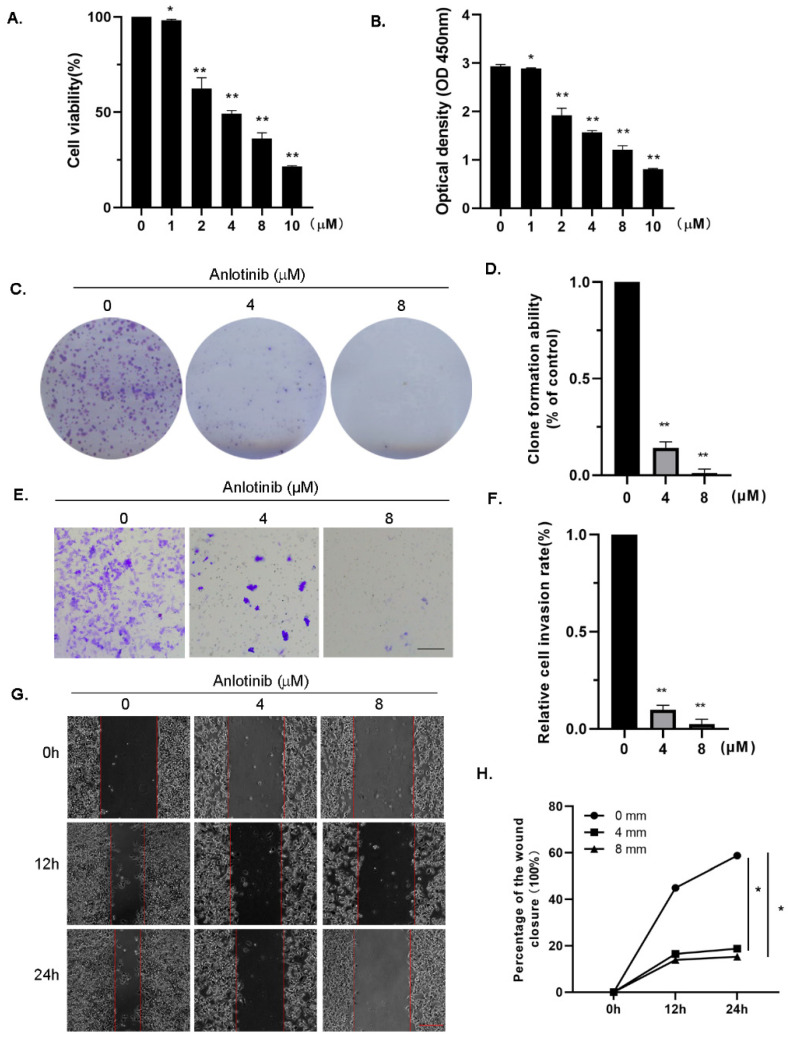
Anlotinib inhibits proliferation, migration, and invasion of ovarian cancer cells in vitro. (**A**,**B**) A2780 cells were treated with different concentrations (0,1, 2, 4, 8, and 10 μM) of anlotinib for 72 h. The cell-counting kit-8 assay was performed. (**C**,**D**) Representative images of colony-formation assay. (**E**–**H**) The effects of anlotinib on cell invasion and migration were measured using wound-healing (**E**,**F**) and transwell assays (**G**,**H**). All experiments were performed in at least three biological replicates, and each biological replicate contained three technical replicates. Student’s *t*-test was used to compare values between control and different treatment groups. * *p* < 0.05, ** *p* < 0.01 vs. control group. OD, optical density. Scale bars = 100 μm.

**Figure 2 molecules-27-08873-f002:**
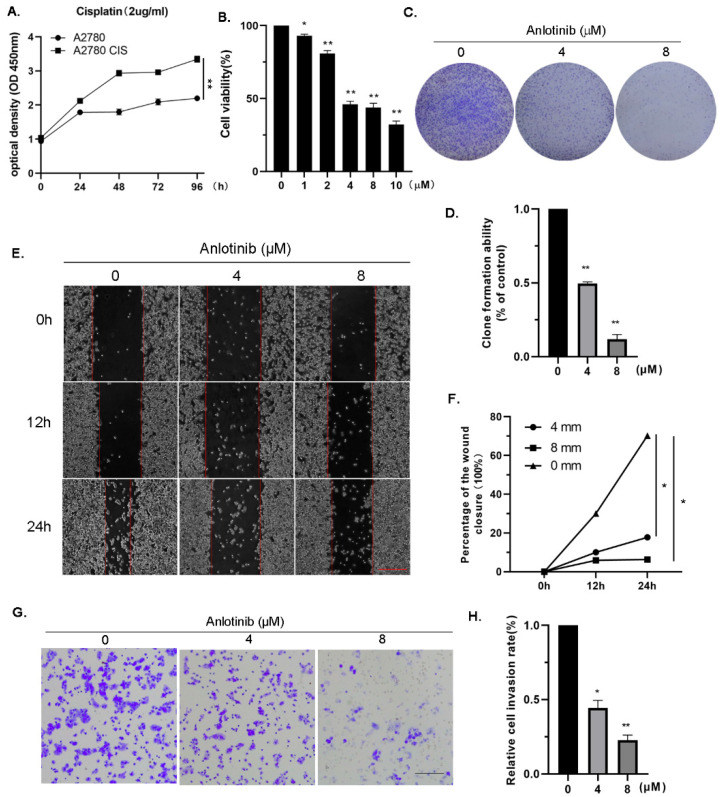
Anlotinib inhibits the proliferation, migration, and invasion of cisplatin-resistant ovarian cancer cells in vitro. (**A**) A2780 and cisplatin-resistant cell line A2780 CIS were treated with cisplatin. Cell viability was measured by CCK-8. (**B**) A2780 CIS cells were treated with different concentrations (0, 1, 2, 4, 8, and 10 μM) of anlotinib for 72 h. Cell viability was measured by CCK8. (**C**,**D**) Representative images of colony-formation assay. (**E**–**G**) The effects of anlotinib on cell invasion and migration were measured using wound-healing (**E**,**F**) and transwell assays (**G**,**H**). All experiments were performed in at least three biological replicates, and each biological replicate contained three technical replicates. Student’s *t*-test was used to compare values between control and different treatment groups. * *p* < 0.05, ** *p* < 0.01 vs. control group. OD, optical density. Scale bars = 100 μm.

**Figure 3 molecules-27-08873-f003:**
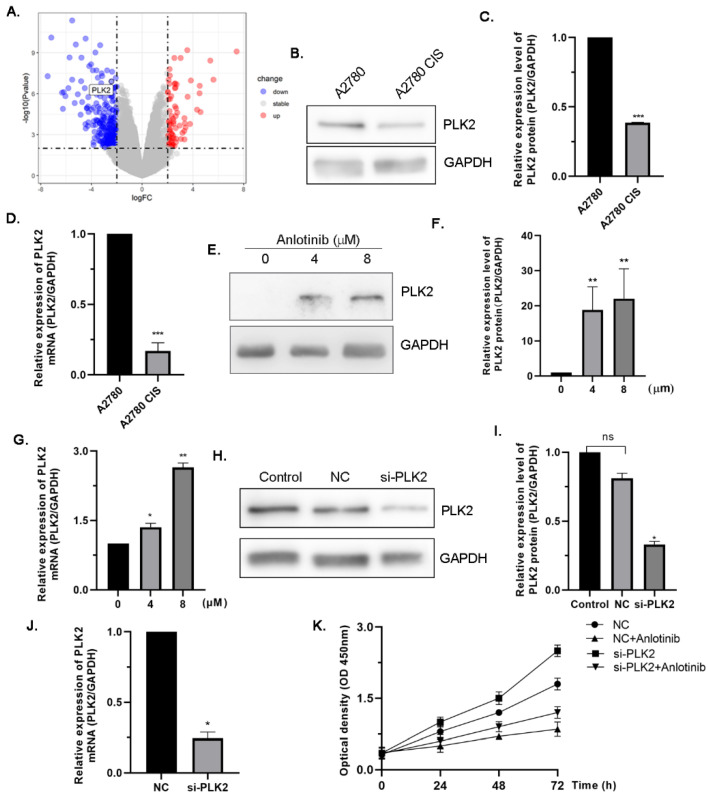
PLK2 is involved in mediating the antiproliferation effect of anlotinib. (**A**) The volcano plot of differentially expressed genes. (**B**,**C**) Western blot analysis of PLK2 in A2780 and A2780 CIS cells (**B**) and the statistical analysis of the change in PLK2 protein (**C**). (**D**) qRT-PCR analysis of PLK2 in A2780 and A2780 CIS cells. (**E**,**F**) Western blot analysis of PLK2 in A2780 CIS cells treated with anlotinib (**E**) and the statistical analysis on the change of PLK2 protein (**F**). (**G**) qRT-PCR analysis of PLK2 in A2780 CIS cells treated with anlotinib. (**H**–**J**) The knockdown efficiency of PLK2 in A2780 CIS cells was assessed via qRT-PCR (**G**) and Western blot (**H**,**I**). (**K**) CCK8 assay detected cell growth under different treatment conditions. All experiments were performed in at least three biological replicates, and each biological replicate contained three technical replicates. Student’s *t*-test was used to compare values between control and different treatment groups. * *p* < 0.05, ** *p* < 0.01 and **** p* < 0.001 vs. control.

**Figure 4 molecules-27-08873-f004:**
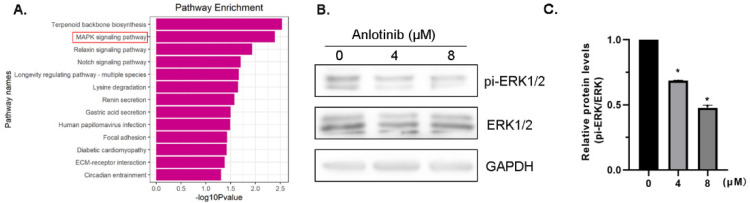
Involvement of the MAPK/ERK signaling pathway in mediating the anti-tumor effect of anlotinib. (**A**) Enrichment analysis of differentially expressed genes. (**B**,**C**) Western blot analysis of ERK1/2 activity in A2780 CIS cells treated with anlotinib (**B**) and the relative quantification of phosphorylated ERK1/2 and total ERK1/2 (**C**). All experiments were performed in at least three biological replicates, and each biological replicate contained three technical replicates. Student’s *t*-test was used to compare values between control and different treatment groups. * *p* < 0.05 indicates a significant difference when compared to the control.

**Figure 5 molecules-27-08873-f005:**
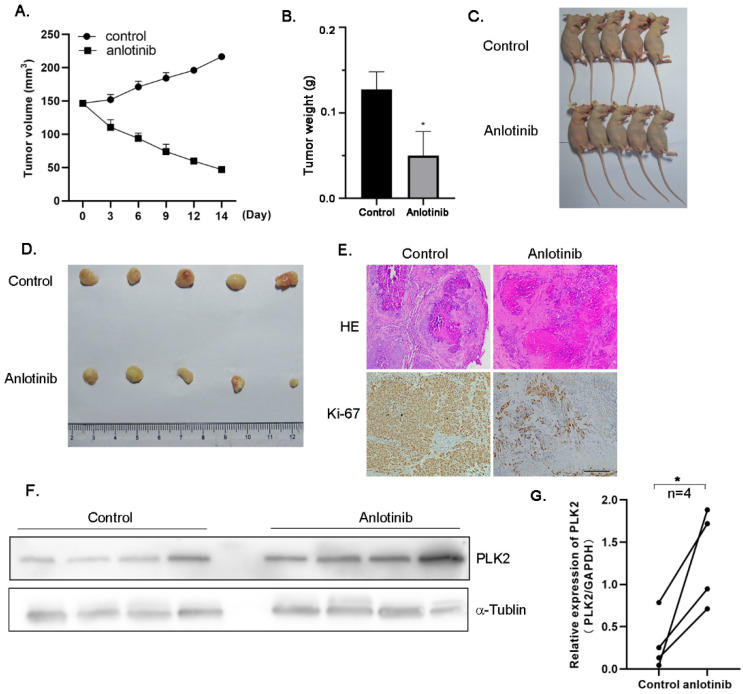
Anlotinib suppresses tumor growth in vivo. Mice (*n* = 10) were intraperitoneally injected with saline or anlotinib (3 mg/kg/day) for 14 days, and the tumor size and bodyweight were monitored every two days. (**A**) Data are plotted as growth curves of the tumor volume and bodyweight. (**B**–**D**) The tumor samples were obtained and weighed after treatment with the drug. (**E**) Representative hematoxylin and eosin and immunohistochemical staining images of Ki67. (**F**,**G**) The expression of PLK2 was quantified via quantitative real-time polymerase chain reaction and Western blotting. All experiments were performed in at least three biological replicates, and each biological replicate contained three technical replicates. Student’s *t*-test was used to compare values between control and different treatment groups. * *p* < 0.05 indicates a significant difference compared to the control. Scale bars = 100 μm.

## Data Availability

Not applicable.

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
