# Peer review of "Anlotinib Exerts Inhibitory Effects against Cisplatin-Resistant Ovarian Cancer In Vitro and In Vivo"

_molecules, 2022, doi:10.3390/molecules27248873_

Round 1

Reviewer 1 Report

The objective of this study was to investigate the inhibitory effect of anlotinib on the pathogenesis of cisplatin-resistant ovarian cancers (OC). The authors demonstrate higher PLK2 expression in OC cell lines on anolitinib treatment, indicating an antitumor effect. Thus, providing evidence of possible mechanism for antitumor effect for cisplatin-resistant OC. This is further supported by in vivo xenograft study. Overall, the study advances our understanding of the mode of action of anolitinib but the authors needs to address few major points.

Major Point:

  1. In the introduction section, the authors needs to elaborate more on what evidence is available for anolitinib to warrant this study.
  2. In Figure 3E, why is the GAPDH expression this low in condition 0 uM? Overall western blotting needs to be redone with an update figure.
  3. The reviewer would like to know the justification for the use of different concentrations of anolitinib in the in vitro (4, 8 uM) and in vivo mice study (3 mg/kg/day).
  4. The manuscript could benefit if the authors had included other tyrosine kinase inhibitors (TKIs) that also inhibits VEGFR2/3 (similar to anolitinib target kinase) in the study design. In addition to validation of the current data these TKIs would be a good control.
  5. Have the authors considered utilizing the current data and incorporating into a PBPK (physiologically based pharmacokinetic) model to predict the outcome in humans?

Author Response

Response to Reviewer 1 Comments

The objective of this study was to investigate the inhibitory effect of anlotinib on the pathogenesis of cisplatin-resistant ovarian cancers (OC). The authors demonstrate higher PLK2 expression in OC cell lines on anolitinib treatment, indicating an antitumor effect. Thus, providing evidence of possible mechanism for antitumor effect for cisplatin-resistant OC. This is further supported by in vivo xenograft study. Overall, the study advances our understanding of the mode of action of anolitinib but the authors needs to address few major points.

Point 1: In the introduction section, the authors needs to elaborate more on what evidence is available for anlotinib to warrant this study.

Response 1: The manuscript has been altered with this suggested change. We have added several lines and new citations in the introduction to further stress the importance of anlotinib in several cancers treatment. Details are now included in the text in lines 57-66.

Point 2: In Figure 3E, why is the GAPDH expression this low in condition 0 uM? Overall western blotting needs to be redone with an updated figure.

Response 2: Western blotting of Figure 3E (F) was redone and statistics were re-quantified

Point 3: The reviewer would like to know the justification for the use of different concentrations of anlotinib in the in vitro (4, 8 uM) and in vivo mice study (3 mg/kg/day).

Response 3: In vitro, the concentrations of 4μM and 8μM were selected because the IC50 of the cell line measured by cck-8 was about 4μM. Additionally, there was literature reported that the concentration of anlotinib was 1.5-6mg/kg when different tumor cells were transplanted into mice [1, 2], and the ovarian cancer cell line SKOV3 showed an obvious effect when the concentration of Anlotinib was 3mg/kg[2]. Therefore, we selected the concentrations mentioned above in vitro and in vivo for experiments.

Point 4: The manuscript could benefit if the authors had included other tyrosine kinase inhibitors (TKIs) that also inhibit VEGFR2/3 (similar to anlotinib target kinase) in the study design. In addition to validation of the current data these TKIs would be a good control.

Response 4: According to previous literature reports, there were some studies on the effect of other tyrosine kinase inhibitors (TKIs) on ovarian cancer, such as Apatinib [3, 4]. There is little research on the effect of TKIs on cisplatin-resistant cells in OC. Therefore, our topic is very innovative. Thank you for your suggestion. We will compare the effects of other TKI and anlotinib on drug-resistant ovarian cancer cell lines in future research.

Point 5: Have the authors considered utilizing the current data and incorporating it into a PBPK (physiologically based pharmacokinetic) model to predict the outcome in humans?

Response 5: Thank you for the suggestion. Physiologically based pharmacokinetic (PBPK) modeling and simulation are used to predict the pharmacokinetic performance of drugs in humans by using preclinical data. It can also explore the effects of various physiologic parameters such as age, ethnicity, or disease status on human pharmacokinetics, as well as guide dose and dose regiment selection and aid drug-drug interaction risk assessment [5]. PBPK modeling has become an integral tool in drug discovery and development. Our research group is very interested in this research direction. But it is different from our group research field, so I know very little about its specific content. If we have the chance to have an in-depth understanding of PBPK in the future, I will apply my existing data to it as you said, and provide some data basis for the subsequent researchers. I think it will be my honor.

Reference:

  1. Song, F., Hu, B., Cheng, J. W., Sun, Y. F., Zhou, K. Q., Wang, P. X., Guo, W., Zhou, J., Fan, J., Chen, Z., and Yang, X. R., Anlotinib suppresses tumor progression via blocking the VEGFR2/PI3K/AKT cascade in intrahepatic cholangiocarcinoma. Cell Death Dis 2020, 11, 573.
  2. Xie, C., Wan, X., Quan, H., Zheng, M., Fu, L., Li, Y., and Lou, L., Preclinical characterization of anlotinib, a highly potent and selective vascular endothelial growth factor receptor-2 inhibitor. Cancer Sci 2018, 109, 1207-1219.
  3. Sun, X., Li, J., Li, Y., Wang, S., and Li, Q., Apatinib, a Novel Tyrosine Kinase Inhibitor, Promotes ROS-Dependent Apoptosis and Autophagy via the Nrf2/HO-1 Pathway in Ovarian Cancer Cells. Oxid Med Cell Longev 2020, 2020, 3145182.
  4. Mallmann-Gottschalk, N., Sax, Y., Kimmig, R., Lang, S., and Brandau, S., EGFR-Specific Tyrosine Kinase Inhibitor Modifies NK Cell-Mediated Antitumoral Activity against Ovarian Cancer Cells. Int J Mol Sci 2019, 20,
  5. Zhuang, X. and Lu, C., PBPK modeling and simulation in drug research and development. Acta Pharm Sin B 2016, 6, 430-440.

Reviewer 2 Report

In this study, Ji et al investigated cisplatin-resistant ovarian cancer. This paper is impressive, and I would like to recommend it for publishing. However, there are some concerns that should be addressed before publishing.

1- You should mention your results in more detail in the Abstract section.

2- In the introduction section, you should provide a better literature review and highlight Anlotinib.

3- Combinate discussion and conclusion; This help to make it easier to follow.

4- And also extend your discussion in detail to help the reader to find Anlotinib exerts inhibitory effects.

5- Improve the cohesion of the paper and remove any ambiguous sentences.

Author Response

Response to Reviewer 2 Comments

In this study, Ji et al investigated cisplatin-resistant ovarian cancer. This paper is impressive, and I would like to recommend it for publishing. However, there are some concerns that should be addressed before publishing.

Point 1: You should mention your results in more detail in the Abstract section.

Response 1: In the abstract, we added an overview of the experimental results (lines 26-32).

Point 2: In the introduction section, you should provide a better literature review and highlight Anlotinib.

Response 2: The manuscript has been altered with this suggested change. Details are now included in the text in lines 57-66.

Point 3: Combinate discussion and conclusion; This help to make it easier to follow.

Response 3: We have reorganized the sentence to stress the anti-cancer effect of anlotinib in discussion section.

Point 4: And also extend your discussion in detail to help the reader to find Anlotinib exerts inhibitory effects.

Response 4: We appreciate this suggestion, and please see above response to Comment #8.

Point 5: Improve the cohesion of the paper and remove any ambiguous sentences.

Response 5: The manuscript has been altered with this suggested change.

Reviewer 3 Report

Anlotinib exerts inhibitory effects against cisplatin-resistant ovarian cancer in vitro and in vivo 

Yurou Ji et al

In this manuscript by Ji et al, the authors investigate the effect of the efficacy of anlotinib, a novel TRK inhibitor, on an ovarian cancer cell line. The authors perform both in vitro and in vivo assays (xenografts) using the cell line A2780 and a cisplatin resistant variant. 

Overall, the study is a decent addition to the scientific literature on anlotinib, and provides novel insights into the mechanism behind anlotinib-induced cell death. Specifically, the authors show that anlotinib therapy induce upregulation of the PLK2 gene, which inhibits cell growth. This inhibition was abolished when PLK2 was inhibited with siRNA.

The language is generally good, but abbreviations must be spelled out first time they are used. This is commonly missing. Not all figure panels are mentioned in the text, and they are not always mentioned in order.

More comments below:

1.     Results section reads as if methods were originally placed above. Each method need a few words of introduction when methods are in the end

2.     Figure 1A is only mentioned in the methods. Either move the figure, or re-word section 2.1 to include mentioning of figure 1A.

3.     Section 2.1, CCK8 not spelled out (what is this)

4.     Section 2.1, Ditto for FCM 

5.     Figure 1 (and all other figures), what are the error bars? Were the experiments performed in replicates? How many replicates? Must be mentioned, either in figure legend or methods

6.     Figure 1 legend, cisplatin and viability misspelled

7.     Figure 1I, Figure 2E, Figure 3C, F, G & I, Figure 4C. Relative levels, shouldn’t the first bar (0µM) reach 1.0? 

8.     Section 2.2, spell out what the CIS cells are. And how are these generated? By the authors?

9.     Section 2.2, attachment 1. What is this? The same as “supplementary data”?

10.  Section 2.3, write out DEG

11.  Section 2.3, what was the rationale for focusing on PLK2? This is not clear, looks like cherry-picking when looking at the figure

12.  Figure 3A, is blue or red higher in what type of cells? 

13.  Line 111, “low” should be “lower”

14.  Line 120, I think you refer to figure 3K, not 4K

15.  Figure 4C not mentioned

16.  Discussion, more reference to literature in general, and to anlotinib in regard to other cancer types would make the work stronger

17.  Dicussion, line 186, “However, relevant…..” this needs re-phrasing, very odd sentence

18.  Conclusions, move section ahead of methods

Author Response

Response to Reviewer 3 Comments

In this manuscript by Ji et al, the authors investigate the effect of the efficacy of anlotinib, a novel TRK inhibitor, on an ovarian cancer cell line. The authors perform both in vitro and in vivo assays (xenografts) using the cell line A2780 and a cisplatin resistant variant. 

Point 1: Overall, the study is a decent addition to the scientific literature on anlotinib, and provides novel insights into the mechanism behind anlotinib-induced cell death. Specifically, the authors show that anlotinib therapy induces upregulation of the PLK2 gene, which inhibits cell growth. This inhibition was abolished when PLK2 was inhibited with siRNA.

Response 1: We appreciate this positive feedback on our manuscript. No change to the manuscript was requested.

Point 2: The language is generally good, but abbreviations must be spelled out the first time they are used. This is commonly missing. Not all figure panels are mentioned in the text, and they are not always mentioned in the order.

Response 2: We note these problems and describe each graph accordingly in this article.

More comments below:

Point 1:Results section reads as if methods were originally placed above. Each method need a few words of introduction when methods are in the end.

Response 1: We note these problems and reorganized the sentence.

Point 2: Figure 1A is only mentioned in the methods. Either move the figure, or re-word section 2.1 to include mentioning of figure 1A.

Response 2: Considering the logic of the result narration, we put Figure 1A in Figure 2A and describe the result in the second part.

Point 3: Section 2.1, CCK8 not spelled out (what is this)

Response 3: The full name of CCK8 is cell counting kit 8, and we have added full spelling when the first time we use the abbreviation.

Point 4: Section 2.1, Ditto for FCM

Response 4: The full name of FCM is flow cytometry, and we added full spelling when the first time we use the abbreviation.

Point 5: Figure 1 (and all other figures), what are the error bars? Were the experiments performed in replicates? How many replicates? Must be mentioned, either in figure legend or methods

Response 5: All experiments were performed in at least three biological replicates, and each biological replicate contained three technical replicates. Differences between groups were analyzed using a two-tailed Student’s t-test and analysis of variance. Values with *P<0.05, and **P<0.01 were considered statistically significant. This statistical method is described in the method part and the legend.

Point 6: Figure 1 legend, cisplatin and viability misspelled

Response 6: The words have been corrected.

Point 7: Figure 1I, Figure 2E, Figure 3C, F, G & I, Figure 4C. Relative levels, shouldn’t the first bar (0µM) reach 1.0?

Response 7: We change Figure 1I, and Figure 2E(F)to a line graph and compare each concentration at each time point. Figure 3C, F&I, and Figure 4C were re-quantified, and the corresponding control group bar was 1. Figure 3G does not need to be modified, which major ticks interval of the Y axis is 0.75.

Point 8: Section 2.2, spell out what the CIS cells are. And how are these generated? By the authors?

Response 8: The full name of CIS cell is the cisplatin-resistant cell, and we have added full spelling when the first time we use the abbreviation. Additionally, the cisplatin-resistant A2780 CIS cell line was a kind gift from Professor Bin Li of Cancer Hospital Chinese Academy of Medical Sciences. We also described it in the Methods section of the article.

Point 9: Section 2.2, attachment 1. What is this? The same as “supplementary data”?

Response 9: attachment 1 refers to the supplementary data, and this error has been corrected in the results section.

Point 10: Section 2.3, write out DEG

Response 10: DEG refers to the differentially expressed genes, and we have added full spelling when the first time we use the abbreviation.

Point 11: Section 2.3, what was the rationale for focusing on PLK2? This is not clear, looks like cherry-picking when looking at the figure

Response 11: First, through differential gene expression analysis, we found that PLK2 was significantly down-regulated in cisplatin-resistant ovarian cancer cells, which was also confirmed by western blot and qRT-PCR. Additionally, PLK2 has been reported to be associated with a variety of cancers and related to drug resistance in ovarian cancer in various literatures. Based on the above evidence, we explored the expression of PLK2 in cisplatin-resistant ovarian cancer cells treated with or without anlotinib.

Point 12: Figure 3A, is blue or red higher in what type of cells?

Response 12: The red section indicates that cisplatin-resistant A2780 cell line has higher gene expression than the A2780 cell line, and the blue section indicates that the cisplatin-resistant A2780 cell line has lower gene expression than the A2780 cell line in Figure 3A.

Point 13: Line 111, “low” should be “lower”

Response 13: The words have been corrected.

Point 14: Line 120, I think you refer to figure 3K, not 4K

Response 14: The referred figure has been corrected.

Point 15: Figure 4C not mentioned

Response 15: We described Figure 4C in line163.

Point 16: Discussion, more reference to literature in general, and to anlotinib in regard to other cancer types would make the work stronger

Response 16: We cited more references regarding anlotinib anti-tumor effects in the discussion section.

Point 17: Dicussion, line 186, “However, relevant…..” this needs re-phrasing, very odd sentence

Response 17: The sentence has been reorganized.

Point 18: Conclusions, move section ahead of methods

Response 18: The section on conclusions was moved to the head of methods as suggested.

Round 2

Reviewer 1 Report

The authors have addressed the reviewer's comments.